# Projecting Future Climate Change-Mediated Impacts in Three Paralytic Shellfish Toxins-Producing Dinoflagellate Species

**DOI:** 10.3390/biology11101424

**Published:** 2022-09-29

**Authors:** Francisco O. Borges, Vanessa M. Lopes, Ana Amorim, Catarina F. Santos, Pedro Reis Costa, Rui Rosa

**Affiliations:** 1MARE—Marine and Environmental Sciences Centre & ARNET—Aquatic Research Network, Faculdade de Ciências, Universidade de Lisboa, 1749-016 Lisboa, Portugal; 2Departamento de Biologia Vegetal, Faculdade de Ciências, Universidade de Lisboa, 1749-016 Lisboa, Portugal; 3Departamento de Biologia Animal, Faculdade de Ciências, Universidade de Lisboa, 1749-016 Lisboa, Portugal; 4Portuguese Institute for the Sea and Atmosphere (IPMA, I.P.), 1749-077 Lisboa, Portugal; 5S2AQUA—Collaborative Laboratory, Association for a Sustainable and Smart Aquaculture, Av. Parque Natural da Ria Formosa s/n, 8700-194 Olhão, Portugal; 6CCMAR—Centre of Marine Sciences, Campus de Gambelas, University of Algarve, 8005-139 Faro, Portugal

**Keywords:** biogeography, climate change, species distribution models, harmful algal blooms, paralytic shellfish poisoning

## Abstract

**Simple Summary:**

Harmful algal blooms present a particular risk for marine ecosystems and human health alike. In this sense, it is important to accurately predict how toxin-producing microalgae could be affected by future climate change. The present study applied species distribution models (SDMs) to project the potential changes in the habitat suitability and distribution of three key paralytic shellfish toxin (PST)-producing dinoflagellate species (i.e., *Alexandrium catenella*, *A. minutum*, and *Gymnodinium catenatum*), up to 2040/50 and 2090/2100, across four different greenhouse gas emission scenarios, and using four abiotic predictors (i.e., sea surface temperature, salinity, current velocity, and bathymetry). In general, considerable contractions were observed for all three species in the lower latitudes of their distribution, together with projected expansions into higher latitudes, particularly in the Northern Hemisphere. This study aims to entice further research on the future biogeographical impacts of climate change in toxin-producing microalgae species while, at the same time, helping to advise the correct environmental management of coastal habitats and ecosystems.

**Abstract:**

Toxin-producing microalgae present a significant environmental risk for ecosystems and human societies when they reach concentrations that affect other aquatic organisms or human health. Harmful algal blooms (HAB) have been linked to mass wildlife die-offs and human food poisoning episodes, and climate change has the potential to alter the frequency, magnitude, and geographical extent of such events. Thus, a framework of species distribution models (SDMs), employing MaxEnt modeling, was used to project changes in habitat suitability and distribution of three key paralytic shellfish toxin (PST)-producing dinoflagellate species (i.e., *Alexandrium catenella*, *A. minutum*, and *Gymnodinium catenatum*), up to 2050 and 2100, across four representative concentration pathway scenarios (RCP-2.6, 4.5, 6.0, and 8.5; CMIP5). Despite slightly different responses at the regional level, the global habitat suitability has decreased for all the species, leading to an overall contraction in their tropical and sub-tropical ranges, while considerable expansions are projected in higher latitudes, particularly in the Northern Hemisphere, suggesting poleward distributional shifts. Such trends were exacerbated with increasing RCP severity. Yet, further research is required, with a greater assemblage of environmental predictors and improved occurrence datasets, to gain a more holistic understanding of the potential impacts of climate change on PST-producing species.

## 1. Introduction

When environmental conditions are conducive to algal growth, there is the potential for a rapid or excessive increase in marine and freshwater populations of phytoplankton [1,2]. While these are natural phenomena that generally do not produce negative impacts on the surrounding habitats [3,4], some instances of exacerbated populational growth (e.g., linked to specific species or to specific abiotic conditions [5]) have the potential to significantly impact ecosystems and human health [6]. Termed harmful algal blooms (HABs), these phenomena have been in the spotlight of environmental management for quite some time [7,8], due to their link to a wide variety of environmental issues. These include mass fish and shellfish die-offs [9,10], the degradation of coastal habitats and ecosystem community structure [11,12], the death of marine mammals and seabirds [13], and outbreaks of human shellfish and finfish poisonings [2,14]. In the marine realm, approximately three hundred of the circa five thousand microalgae species (c. 6%) are known to have the potential to generate HAB events under favorable conditions [5,15]. In short, these include high-biomass and toxin-producing microalgae species, with each leading to slightly different environmental impacts: the former being linked to significant harmful impacts on marine communities and food webs (e.g., by inducing severe anoxia [16,17]), while the latter produce toxins with varying degrees of toxicity for marine species and other organisms in the trophic web, e.g., birds and humans [16,18].

More than one hundred marine planktonic species of microalgae are known to produce toxins, the majority belonging to the Phylum Dinoflagellata [5,19,20,21]. Paralytic shellfish poisoning (PSP) is one of the most widespread HAB-related shellfish poisoning syndromes [18]. This syndrome is caused by paralytic shellfish toxins (PSTs), a suite of saxitoxin derivatives that are considered one of the most significant toxin-related environmental hazards [18,22,23]. The impacts of PST include changes to the marine trophic structure and an increased mortality of marine mammals, fish, and seabirds [18,23], a loss of seafood resources, impairment of leisure ecosystem services, and potential human mortality [24]. Several dinoflagellate species from the *Alexandrium* genus and *Gymnodinium catenatum* Graham are some of the most well-known PST-producing species. In the *Alexandrium* genus, species such as *A. catenella* (Whedon & Kofoid) Balech [25] and *A. minutum* Halim have been studied in detail [23,26,27]. Both have a wide geographical distribution and can colonize a vast array of habitats and hydrogeographic regimes, due to their remarkable resilience and adaptability. However, *A. minutum* is mostly confined to estuaries and coastal lagoons, not being found in shelf waters, as are *A. catenella* and *G. catenatum*. Water temperature and salinity (through river flow/freshwater input) are known to be some of the main factors impacting the magnitude, growth rate, and length of *Alexandrium* spp. blooms [28,29,30,31]. The chain-forming *Gymnodinium catenatum* is the only species within the genus *Gymnodinium* producing PST in coastal waters worldwide [32]. This species was first described from the Gulf of California, and has a wide geographical distribution, being found across the Americas, Europe, and Oceania, as well as North Africa and the western Pacific [33,34,35]. Reports of new occurrences since the 1970s have been linked to ballast water dispersal and global warming, among other causes [36,37,38,39]. 

During the last few decades, there has been an increasing understanding of the potential for climate change effects on the ocean to alter the frequency, magnitude, and geographical spread of both toxic and non-toxic HAB events [30,40]. The rising atmospheric concentration of carbon dioxide (CO_2_) during the past centuries [41,42], largely linked to anthropogenic activities, has created significant changes in the Earth’s climate and to the global ocean [43]. Changing oceanic chemistry, decreasing pH levels, rising sea surface temperatures [44,45], increased stratification and slowed ocean circulation, and oceanic oxygen loss [46,47] are, simultaneously, consequences and drivers of oceanic climate change, which are set to yield a vast array of impacts on marine ecosystems [48,49,50]. Climate change experts have been following the present and potential future ecological and socioeconomic impacts of climate change for some time now, producing a series of reports featuring projections based on total greenhouse gas concentrations and environmental actions, which set the projected average changes in environmental conditions globally until the end of the century [51]. In one of the latest iterations of these reports, the experts utilized four greenhouse gas concentration trajectories in climate modeling—termed Representative Concentration Pathway (RCP))—which are labelled after a possible range of radiative forcing values for the end of the century [52]. 

While there is still a lack of empirical support that the occurrence, toxicity, and risk of HABs globally exhibit an increasing trend [53], at the regional level, there are reports of increasing numbers of PSP outbreaks in the past few years, as bloom-favorable conditions appear to become increasingly more frequent [17,40]. Warmer waters are already known to be affecting bloom dynamics [54]. For example, anomalously warm sea surface temperatures have been shown to lead to increased closures of shellfish harvesting, due to PSTs produced by *Alexandrium* species in some regions of the world [55]. Salinity and current velocities can also influence HAB dynamics, by influencing occurring species, algal growth, and the potential accumulation and dispersal of blooms (e.g., blooms can also occur by population convergence) [56,57]. Notwithstanding, it is likely that warming may induce differential effects, depending on the species and region in question, by stimulating the growth rates of organisms inhabiting the poleward limits of their thermal habitat, but at the same time reducing growth in regions that become too warm to support growth [38,58]. This has the potential to significantly alter the composition of phytoplankton communities, by changing the occurrence and geographic spread of bloom-forming species [2,17,39], while at the same time affecting the phenology of such species [59] and widening the window of optimal conditions for blooms to develop [40,60]. 

In this context, efforts to try to model potential changes in the distribution of HAB species are particularly important [38]. Indeed, combining biogeographic knowledge in climate change impact studies and modeling allows the prediction of the potential future impacts of environmental changes on biodiversity and ecosystem health [16,61]. In this sense, species distribution models (SDMs) are a highly useful tool. Having seen a steep rise in their development and use in the past decades [62], SDMs offer a suitable framework for predicting changes to species distributions, regarding large species assemblages and across vast geographical spaces [63]. This modeling tool has been increasingly used to project the potential effects of climate change on species’ and communities’ distributions [64,65,66,67,68], and SDMs are considered an effective way to address some of the questions regarding the effects of climate change on biodiversity [69]. These models have already been used to project changes in the distribution of phytoplankton [70], and specifically of HAB species [40,71], under marine climate change, although mostly at regional scales.

The aim of the present study was to evaluate the potential worldwide biogeographical impacts of future oceanic climate change on the distribution of three key PST-producing marine dinoflagellate species: (i) *Alexandrium catenella*; (ii) *Alexandrium minutum*; and (iii) *Gymnodinium catenatum*. Specifically, this study implemented an SDM workflow using MaxEnt modeling, to model the present-time habitat suitability and species occurrence distribution, and project these into two future time periods (i.e., 2050 and 2100), across four RCP scenarios (RCP-2.6, 4.5, 6.0, and 8.5, CMIP5). 

## 2. Materials and Methods

### 2.1. Data Collection and Curation

Species occurrence data were collected from the Global Biodiversity Information Facility (GBIF) [GBIF.org (14 February 2022)]. A specific set of filters was used to limit the data retrieved in each dataset, positively selecting for “Human Observation” and “Preserved Specimen” as the Basis of Record, and for data that is georeferenced, while filtering out duplicate observations, improper datum conversion points, missing (NA) values in either longitude or latitude, and rounded latitude/longitude coordinates. The compiled dataset was then curated using RStudio [72]. For each species, the occurrences were restricted to those occurring from the surface to the mean maximum depth of the continental shelf (~200 m) [40,73]. To achieve this, and since GBIF does not include depth data for most species, the occurrence data was converted into a spatial polygon object, which was then used to extract the depth values at each occurrence’s coordinates (using the function ‘extract’, from the package ‘raster’) from a bathymetry raster layer obtained from Ocean Climate Layers for Marine Spatial Ecology (MARSPEC) [74]. The spatial polygon objects were then converted back into data frames and merged with the geo-referenced depths, and each species’ data frame was subset to exclude depths greater than 200 m. A second clipping of the occurrence data aimed at removing data occurring on land by removing all the points outside a shapefile from all the ocean bodies, downloaded from Natural Earth Data [https://www.naturalearthdata.com (14 February 2022)]. This restriction procedure of the species occurrences was performed since SDMs must ideally restrict the model calibration to accessible areas [75]. The post-curation number of valid entries per species for the analysis is provided in Table 1. The curated occurrence dataset, together with the plotted dataset occurrences for each species and the R script used in this section, are presented in the Appendix A (See “data_curation.R; and the “Curated_species_datasets” and “species_occurrence_plots” folders). 

### 2.2. Predictor Variables

The predictor variables used in this study included three oceanographic variables, i.e., the mean, max, min, and range layers of the sea surface temperature (SST), salinity, and current velocity; and one topographic variable, i.e., bathymetry. The choice of the oceanographic variables was primarily based on the availability of environmental predictors projected for both present (i.e., 2000–2014) and future periods (i.e., 2040–2050 and 2090–2100), and RCP scenarios of interest. The four main RCP scenarios include RCP-2.6, which requires CO_2_ emissions to decline from 2020 to reach zero in 2100, and projects the global temperature rise to stay below 2 °C by the end of the century. RCP-4.5 is an intermediate scenario, with emissions peaking by 2045, resulting in an increase in approximately 2 °C to 3 °C by the end of the century. RCP-6.0 is a high greenhouse gas emissions scenario, with total radiative forcing stabilizing by the year 2100, resulting in a potential temperature increase of 3–4 °C. Finally, RCP-8.5 is considered the “worst-case scenario”, based on continued increasing emissions throughout the 21st century, leading to potential increases of over 4 °C [52,76]. These scenarios have been widely used by researchers to project the potential future impacts on species and ecosystems worldwide. These variables were obtained from Bio-ORACLE, which offers global geophysical, biotic, and climate layers at a common spatial resolution (5 arcmin) and a uniform landmask [77,78]. In turn, bathymetry (retrieved from MARSPEC) was used for occurrence data clipping (as described above), and as a predictor variable, to incorporate total water column height in the spatial analysis. 

The following procedures were conducted using the package megaSDM [79], which allows the efficient synthesis of SDMs for several species, time periods, and climate scenarios, presenting a simple and intuitive workflow. Since megaSDM relies on MaxEnt modeling, all the predictor raster layers were reprojected to an equal-area projection (i.e., specifically, the cylindrical equal-area projection: “+proj = cea + lat_ts = 0 + lon_0 = 0 + x_0 = 0 + y_0 = 0 + datum = WGS84 + no_defs”, using the nearest neighbour method). This was performed since conventional non-equal area projections have grids that vary in their area when moving away from the equator. MaxEnt randomly samples cells from the available geographic space, implicitly assuming cells of equal area in the entire extent of each predictor layer, which bears the need for reprojecting the layers a priori [80]. At this point, the functions ‘TrainStudyEnv’ and ‘PredictEnv’ were employed to manipulate and standardize the present and future time periods’ input environmental data, by re-projecting, clipping, and resampling the raster predictors [79]. The former function also defines the training area, where the occurrence and background points are located, the study area where the model will be projected, and the habitat suitability predicted. 

### 2.3. Modeling

The megaSDM package employs a series of measures to mitigate the inherent bias existing in collected or downloaded occurrence data [81,82], and which decreases the overall accuracy of SDMs [81,83,84]. In short, the package mitigates environmental and spatial biases within occurrence data by environmentally filtering the occurrence data [84], i.e., dividing the environmental data into a given number of bins (n = 25 in the present study) and selecting one point from each unique combination of bins, resulting in a subset of occurrence points filtered by the environment [79]. This method allows the removal of clustered or oversampled records, while still maintaining the range of environments in which a species was found [84]. The script for the SDM analysis is present in the Appendix A (See “sdm_analysis.R”).

Given that the species occurrence data frames featured only presence data, there was a need to generate background points for each species, describing the environmental conditions of the training area. This was performed since SDMs must account for absence data, either with the incorporation of true data or pseudo-absence data [75]. To achieve this, the package utilizes a ‘combined’ method, which uses both random and spatially constrained sampling of the background points [79]. In summary, 50% of the set number of background points (n = 1000 per species) was randomly sampled from the entire study area [85], while the remaining 50% was sampled from within buffers created around each true occurrence point, with the radius of each buffer being proportional to the 95% quantile of the distance to the nearest neighbor for each point [79]. The use of this combined background point generation method allows the reduction of environmental suitability overestimation by the model, in regions with greater densities of occurrence points in more easily sampled areas (i.e., spatial bias), which the random method by itself does not consider [86,87], while also reducing the susceptibility of extreme extrapolation errors and overfitting induced by using purely spatially constrained methods [88]. During generation, the background points were also environmentally filtered, in a similar way to that of the occurrence data, creating an even spread across the available environmental space, while retaining its spatial weighting [79]. The background point dataset is present in the Appendix A (See “Backgrounds” folder).

The present habitat suitability and distribution of each species were estimated using the MaxEnt modeling technique, which employs maximum entropy methods and machine learning and has a predictive performance that is consistently competitive with the highest-performing methods [63,88], and performs particularly well when dealing with presence-only species records [89]. For this purpose, the megaSDM package relies solely on MaxEnt modeling instead of resorting to create ensembles of several different modeling methods, which introduces uncertainty in the consensus model [88]. Notwithstanding, to generate statistically rigorous models, the package allows for replication with the subsequent ensemble [79]. In this sense, a replicate number of five was used, meaning the MaxEnt algorithm ran five times per species, each time with a different subset of occurrence points. The evaluation of each model replicate was performed in the ‘MaxEntProj’ function of the package, which takes the area under the curve (AUC) values for each replicate and compares them to null models, where the multiple replicates of the occurrence points are placed randomly in the training area and compared to a random subset of the null data [79,90]. The function then removed all the models with a validation AUC value lower than 0.70. The MaxEntProj function then projected all the models onto the current and future environments, across all the RCP scenarios, and created the ensemble of all the replicate maps, using the median value of each pixel, and reducing the effect of the outliers [79,91]. The evaluation plots and tables for each species are present in the Appendix A (See “Evaluation” folder). 

From the ensembled models, the habitat suitability maps were obtained. To obtain the binary maps of the probability of occurrence (0 or 1), a threshold value was employed on the continuous habitat suitability maps—the mean model TSS criteria of model evaluation, ‘maximum test sensitivity and specificity’ logistic threshold, which maximizes the specificity and sensitivity of the receiver operating curve (ROC) and is particularly effective in presence-only data [92]. The ensembles of habitat suitability for each species, time period, and RCP scenario, as well as the respective binary maps of probability of occurrence, are supplied in the Appendix A (See “Maps” folder). 

### 2.4. Post-Analysis

The post-analysis phase encompassed a visual and quantitative analysis of the potential changes in habitat suitability and species distributions across space and time. First, the changes in the global latitudinal distribution were assessed by plotting the latitudinal centroids (i.e., the arithmetic mean latitude for the species-occupied pixels) for each species, time, and RCP scenario. Then, the latitudinal trends in the habitat suitability were also obtained, by first converting each habitat suitability ensemble into a matrix form and calculating the mean value of each row. The resulting vector of the mean habitat suitability for the present-day was then subtracted from the 2050 and 2100 vectors, thus obtaining the changes in the mean habitat suitability by latitude, which were then plotted along the latitudinal (y) axis to visualize the potential latitudinal shifts across time and RCP scenario [79]. Afterwards, to evaluate changes in the species’ distribution, the binary maps of probability of occurrence were processed in the ‘createTimeMaps’ function of the megaSDM package. These maps allow the visualization of both unidirectional range shifts and transitory fluctuations, i.e., range contraction followed by expansion, or vice-versa [79,93]. In short, the function takes the predicted present-time distribution of each species and subtracts it from the projected future distribution in 2050 and 2100, to obtain a map of the distributional changes across time for each species across each RCP scenario. The time maps for each species and RCP scenario are supplied in the Appendix A (See “Maps” folder). The post-analysis scripts can be found in the Appendix A (See “Post_analysis” folder).

## 3. Results

Concerning the model evaluation, the results indicate that the variable importance in the MaxEnt model runs was heavily skewed towards the bathymetry predictor, with this layer contributing over 60% to the model predictions. At the same time, the dynamic environmental variables (i.e., current velocity, salinity, and temperature), exhibited considerably smaller contributions: specifically, the second-highest contributing variable for each species was the maximum temperature (~10%; for *A. minutum*), mean temperature (~10%; for *A. catenella*), and the salinity range (~8–10%; for *G. catenatum*). Regarding the model results, there is a common trend between the three species concerning the differences in the centroid of latitudinal distribution (Figure 1), with all the species exhibiting a vertical shift in the centroid position between the present-day and the year 2100, towards northern latitudes—except for *A. catenella* under RCP-2.6 (Figure 1, left). For *A. catenella* and *A. minutum*, however, there is an apparent southward shift of the centroids between the present-day and the middle of the century, before increasing until 2100, with the centroid values decreasing below the present-day baselines. This is the case for RCP-4.5, 6.0, and 8.5 in the case of *A. catenella* (Figure 1 left), and RCP-2.6 and 6.0 for *A. minutum* (Figure 1, middle). Concerning the northward shift until the end of the century, the difference in the centroid position between the present-day and 2100 increases alongside the RCP severity for all species. Finally, only *G. catenatum* exhibits a continuously northward shift in the centroids of its distribution, with both 2050 and 2100 values staying above the present-day baseline (Figure 1, right). 

In the next section, the plots showing the latitudinal trends in the mean habitat suitability for the three studied species (top to bottom: RCP-2.6, 4.5, 6.0 and 8.5; left to right: 2050, 2100), and the maps of the difference in the projected binary occurrence distribution across time (top to bottom: RCP-2.6, 4.5, 6.0, and 8.5) are presented. Concerning the changes in the mean latitudinal habitat suitability for *A. catenella*, there is a common trend across all the RCP scenarios, in which the habitat suitability at higher latitudes (both in the Northern and Southern Hemispheres) is projected to increase until 2100 (Figure 2). At the same time, there is a decreasing habitat suitability over most tropical and subtropical areas near the equator, with a relatively small loss in suitability being projected until the end of the century. This increase until 2100 for the higher latitudes is particularly exacerbated with increasing RCP severity, mainly in the Northern Hemisphere. These changes in the mean habitat suitability are followed by changes in the occurrence distribution (Figure 3). These are relatively sparse for RCP-2.6, with losses in the occurrence distribution being projected mostly in subtropical areas—from the Black Sea to the coastlines in the Adriatic and Aegean Seas, some areas in the North African coastline of the Mediterranean, and in the Yellow Sea, in the north-western Pacific Ocean; as well as tropical regions, including the southern Caribbean Sea, near Aruba, the coastlines of the Andaman Sea in the eastern Indian ocean, the Gulf of Thailand, and in the limits of its projected distribution in the Seas of Timor and Arafura, near Australia. In this scenario, the models predicted relatively few occurrences of areas of transitory fluctuations—particularly expansions followed by contractions. These are mainly located across the tropical belt, in areas such as the Gulf of Mexico, the Seychelles islands, the Palk Strait between India and Sri Lanka, and the Gabon coastline in West Africa. The projected expansions for 2100 are mostly limited to the higher latitudes in both hemispheres. For example, expansion is projected across the Great Australian Bay and the Bass Strait in Oceania; the southern tip of the South American continent in the Southern Hemisphere; the southeastern and western coastlines of Canada; and several areas in the eastern Atlantic Ocean (e.g., the Gulf of Biscay, the English Channel, and the coastline of the United Kingdom), together with the southern coastlines of the North Sea. Lastly, some expansions are also projected for the area of the Yellow Sea. In RCP-4.5, the pattern is quite similar: overall, losses appear to increase by 2050, mainly in the Mediterranean Sea, the southwestern Atlantic Ocean, and Southeast Asia, while the projected gains in occurrence by 2050 and 2100 become more evident towards higher latitudes, such as in the Gulf of Alaska and southern New Zealand, expanding in the southern regions of the North Sea. Regarding RCP-6.0, there is a considerable expansion, followed by a contraction, projected around the area of the Gulf of Thailand and in the sea east of Malaysia and the Island of Sumatra. Projected decreases already present in RCP-4.5 in the regions of Southeast Asia and the Mediterranean Sea remain, as well as those in the southern Caribbean Sea. The gains in occurrence distribution occur in the same areas as predicted for the previous RCP scenarios, despite featuring a slight but evident expansion towards higher latitudes, and further away from the shorelines. Finally, in the ‘high emissions scenario’, RCP-8.5, while the losses in distribution are similar to those in the prior RCP scenarios, albeit with some minor increases (e.g., in the region of the Yucatán Peninsula), there is a considerable increase in the areas of projected expansion by 2100. Specifically, large areas of expansion are projected for the southern extensions of South America (i.e., the southern Chilean and Argentinian coastlines), as well as in the Bass Strait, and in southern New Zealand in the Southern Hemisphere. In the northern latitudes, the coastlines of the Gulf of Alaska and the areas around Nova Scotia exhibit considerable expansions. The eastern Atlantic Ocean and the North Sea constitute areas of considerable gains in distribution. Two more instances worth mentioning are the Black Sea area and the north-western Pacific (i.e., in the areas of the Yellow Sea, Eastern China Sea, and the Sea of Japan), which exhibit considerable gains in *A. catenella* distribution until the end of the century.

Regarding *A. minutum*, the mean latitudinal habitat suitability exhibits a considerable increase in the northernmost latitudes of its distribution, mainly in the polar and subpolar regions (Figure 4). At the same time, the remaining latitudinal profile is dominated by a decreasing habitat suitability, with small losses in the Southern Hemisphere, and in the tropical and subtropical regions surrounding the equator. However, in the temperate latitudes in the Northern Hemisphere, there is a considerable loss of habitat suitability regarding the present-day baseline. This pattern is, similarly to *A. catenella*, exacerbated by the increasing RCP severity. The projected changes in the occurrence distribution follow these trends (Figure 5). Indeed, most of the contractions projected for the distribution of *A. minutum* are predicted to occur at the lower latitudes of its predicted range. In RCP-2.6, a loss of distribution is predicted for the Adriatic and Aegean Seas and the south-eastern Mediterranean coastlines; the eastern and western shores of Australia; and the southern Japanese shores, together with Korea and parts of the Yellow Sea region. In the western Atlantic Ocean, losses are predicted in sparse locations in the Gulf of Mexico, Southern Brazil, and along the eastern coastline of the United States and Canada. In this scenario, there is also a projected expansion in this species’ distribution in the Northern Hemisphere, particularly in the latitudes of the Arctic Circle (e.g., Barents Sea, Norway Sea), with sparse expansion also occurring on the coastlines of the Bering Sea and the Okhotsk Sea in the northern Pacific Ocean. The increasing RCP severity has exacerbated the gains and losses, mostly in the regions mentioned above. A considerable contraction is observed on the Mediterranean coastlines and along the north-western margin of the Atlantic Ocean, as well as in eastern and western Australia, and in the region of the Yellow Sea and southern Japan. The distribution loss also becomes evident in certain regions of the western Canadian coastline, and along specific regions of the western American coastline (e.g., in Mexico, Ecuador, and Peru) and the south-eastern margin of the Atlantic (e.g., the coastlines of Brazil and Uruguay). Concerning the areas of distribution expansion, these grow particularly in the higher latitudes, with the Bering Sea, the Sea of Barents, and the Arctic Ocean exhibiting most of the clear gains.

Lastly, the mean latitudinal habitat suitability for *G. catenatum* is projected to considerably change along the entire latitudinal profile (Figure 6). Indeed, for all the RCP scenarios, there is a considerable loss in suitability being projected for most tropical and subtropical regions, and also in the temperate regions of the Northern Hemisphere. Gains are projected to occur mainly in the subpolar and polar latitudes of the Northern Hemisphere, despite relative increases at subpolar latitudes in the Southern Hemisphere. Finally, this pattern is also exacerbated with the increasing RCP severity, with the greatest differences occurring by 2100 for RCP-6.0 and 8.5, when there is a considerable decrease in the habitat suitability over the tropics and most of the temperate latitudes, and a significant increase in the subpolar and northern temperate latitudes, which increases until the year 2100, and a more moderate increase, stabilizing in 2050, at the higher latitudes of the Southern Hemisphere. This pattern is closely followed by the projected changes in the occurrence distribution (Figure 7). Indeed, in RCP-2.6, the expansion is almost non-existent, apart from some sparse areas in the uppermost northward latitudes (i.e., the coastlines of Japan and Russia in the Okhotsk Sea). In this scenario, transitory fluctuations, namely contractions followed by expansions, are observed in the north-west Atlantic Ocean, to eastern Canada on the North American coastline, and in the north-west Pacific Ocean, in the Yellow Sea. Concerning the projected contractions in this species’ distribution, most occur in the interface between the eastern Indian and western Pacific Oceans, from the areas of the Andaman Sea and the Gulf of Thailand to the Java Sea, but also along the eastern Indian coastline and the southern entrance to the Red Sea, and in the region of the Maldives. In RCP-4.5, there is some increase in the areas of distribution loss, albeit limited. These occur mainly in the Indian Ocean, in the Seychelles archipelago, the Persian Gulf and the Gulf of Oman, and across the north-western coastline of Myanmar. However, a distribution contraction is also projected in the Seas of Timor and Arafura; in Oceania; on the coastlines of Brazil and the Gulf of Guinea in the Atlantic Ocean; and along the Pacific coastline of Mexico. Again, the expansion is limited, with most of the increases occurring in the Gulf of Alaska and in the Yellow Sea, as well as other areas already highlighted in RCP-2.6. In RCP-6.0 and RCP-8.5, however, the areas of distribution loss become considerably larger and more widespread, while the areas of distribution expansion also gain size, but remain relatively restricted to the same regions already identified in the other scenarios. Specifically, the areas of projected distribution loss expand in South and Central America; in most of northern Brazil; the Gulf of the Caribbean (e.g., the coastlines of Colombia, Jamaica, and Cuba); the Guianas; and both the Pacific and Atlantic coastlines of Mexico. In the eastern Atlantic Ocean, the situation is approximately the same, with losses in the Gulf of Guinea occurring from Sierra Leone to Gabon, and in the Indian Ocean, as well (with most of the western Indian coastline also included, in RCP-8.5). The greatest contraction in the occurrence distribution of *G. catenatum* occurs in Southeast Asia, from the Gulf of Thailand and the South China Sea westwards into the sea around the Philippines, and southward, encompassing most of the Arafura and Timor Seas, and the northern Australian coastline. Finally, concerning the expansion, it is comparatively exacerbated in RCP-6.0 and 8.5, with considerable gains in the distribution extent occurring along the Bering Sea and the Gulf of Alaska; in the northernmost latitudes of the Pacific Ocean; and also in the Barents Sea and the northern coastline of Iceland near the Arctic Circle. Some restricted expansion also occurs in the Falkland Islands and in the French Southern Territories in the Southern Hemisphere (i.e., near the Southern Ocean), and in the Sea of Japan and Okhotsk Sea in the north-west Pacific Ocean.

## 4. Discussion

The aim of the present study was to conduct a global analysis of the potential biogeographical response of three key PST-producing dinoflagellate species to the effects of climate change in the oceans, until the end of the century. In general, the models were able to correctly predict the known present-day distribution of each species. Overall, the three species exhibited a northward displacement of their centroid of distribution by 2100, regardless of the RCP scenario (with the single exception of *A. catenella* for RCP-2.6). This suggests that these species might undergo a poleward shift in the near future, which is more or less exacerbated, depending on the severity of the emissions scenario (Figure 1). Notwithstanding this shared trend, the latitudinal profiles in the change of the mean habitat suitability and the time maps suggest the existence of differential spatial patterns for each species. Regarding the variable importance, bathymetry was consistently the most important variable to the models (with values of approximately 60% and over), which is to be expected when dealing with neritic species, and with occurrence records that are mostly from coastal monitoring programs. In this sense, the second and third variables (i.e., maximum temperature and salinity range) were considered the most ecologically relevant to the three species.

Concerning *A. catenella*, the predictions suggest a widespread, global distribution. Indeed, the model was quite accurate in its predictions, in the north-western Pacific Ocean [94,95,96] and the south-western Pacific Ocean [29,97,98,99]; South Africa [100]; South America [101,102]; the Mediterranean Sea [103,104]; and on the eastern shores of North America [105]; as well as the West Coast [106,107]. An overprediction did occur for certain regions, where no major blooms or occurrences have been recorded, as in some regions of the African continent (although the absence of records could be linked to the very small number of studies); on the Atlantic coast of the Iberian Peninsula (although in this region it could occur in very low concentrations that escape the monitoring programs and phytoplankton studies); and in other European areas, where it is only occasionally reported. Regarding the future projections, under RCP-2.6, this species exhibited a latitudinal decrease in its centroid by the year 2100 (Figure 1), which, although seemingly contradictory, is explained by the relatively high contribution of the increase in the mean habitat suitability between 2050 and 2100 at the higher latitudes of the Southern Hemisphere, mainly in the regions of South Australia, New Zealand, and Argentina (Figure 2). This, together with a moderate increase at the mid-latitudes in the Northern Hemisphere, led to the mean centroid decreasing until the end of the century, compared to the present-day baseline (Figure 1). However, regarding the other scenarios, the considerable increase in habitat suitability at the northernmost temperate latitudes in the Northern Hemisphere (Figure 2) led to a considerable expansion of this species’ distribution, particularly under the more severe RCP scenarios (Figure 3), as seen by the considerable shift north of the centroid (Figure 1). In recent years, this species has been shown to occur in the surface waters of the Alaskan Arctic region and develop recurrent local blooms, as evidenced by the presence of extensive resting seed-beds in the bottom sediments [107,108], which the present models did not predict. For this region, the recent literature has shown that warming has the potential to facilitate the range expansion of this species in this region (specifically in the Chukchi Sea), leading to an increased frequency, magnitude, and duration of *A. catenella* blooms [108,109]. In the Mediterranean Sea and the Chilean coast, the expanding blooms could also be linked to temperature changes, alongside other anthropogenic factors [101,104]. Notwithstanding, laboratory studies have shown that certain populations of *A. catenella* might suffer reduced growth rates at elevated future temperatures (e.g., in the Gulf of Maine, on the East Coast of North America [110]).

For *A. minutum*, the models were able to produce an overall satisfactory representation of the present-day distribution (Figure 5). Indeed, this species is considered to have a worldwide distribution [111,112,113], despite its discrete occurrence [114]. This species’ main area of occurrence is located in the Northern Hemisphere, namely in northern Europe, the Iberian Peninsula, and the Mediterranean [112,114]. Odd occurrences of *A. minutum* have also been described from the Azores [31], and elsewhere around the world [112]. It is necessary to note that the species’ environmental affinities (e.g., low salinity, water stratification, and freshwater nutrient runoff [114]) limit this species’ occurrence to sheltered coastal areas, such as estuaries and coastal lagoons [112], which may prevent an accurate description of the present-day distribution by global oceanic models. Notwithstanding, the model was able to accurately describe most of the known occurrences. Regarding the future projections, by 2050 the considerable and sustained loss of the mean habitat suitability over most of the species’ latitudinal profile (Figure 4) (except for the higher northern latitudes) led to the decreasing latitude of the mean centroid of distribution (Figure 1) in RCP-2.6 and 6.0. However, by 2100, the increasing habitat suitability led to a shift of the centroid northwards, opening new geographic areas for the species to expand into. However, *A. minutum* showed some differences in its response to the future climate, compared to *A. catenella*. First, while *A. catenella* did not exhibit considerable losses in its global distribution over time and across RCPs (Figure 3), *A. minutum* exhibited considerable contractions in its distribution in areas, such as the Mediterranean Sea, the Yellow Sea, and the Sea of Japan; and in the temperate latitudes of the Pacific and Atlantic coastlines of the American continents; as well as both the western and eastern coastlines of Australia (Figure 5). This suggests that *A. minutum* could potentially suffer more deleterious impacts of marine climate change when compared to other species from the same genus. However, *A. minutum* exhibits the same poleward distributional shift effect and is projected to expand into the Arctic Ocean (Figure 1 and Figure 4), where it could potentially benefit from the effect of lower salinity [114,115]. If these projections are accurate, it would mean that *A. minutum* has the potential to be one of many other species of marine phytoplankton predicted to further move into this region of the world’s oceans, with the potential to induce severe changes to established marine communities in the Arctic, as well as to potentially increase the risk of toxic HAB events [70]. 

Concerning *G. catenatum*, the SDMs were also able to produce a good global representation of the current distribution of this species (Figure 7). For the American continents, *G. catenatum* was predicted to occur in a wide range of regions where its presence has been described. Specifically, it was predicted to occur from the southernmost latitudes in the western Atlantic [25,37,116,117] to the Gulf of Mexico [116]; across the Atlantic from its northernmost distribution limit on the west coast of Iberia [34] to Angola [34,118]; as well as the Mediterranean Sea [119,120]. In the region of the Black Sea, however, the models did not accurately predict this species’ known occurrence [119]. The predictions for the Indian Ocean were also accurate [121,122,123,124], as were those for Oceania [125,126,127]. In the Pacific Ocean, this species was predicted across Southeast Asia, from Thailand to the Yellow Sea [128,129,130,131,132,133,134,135] and northward into the Russian islands north of Japan, together with the continental shores of Russia near the Okhotsk Sea [136]. While contemporary records of *G. catenatum* north of the Iberian Peninsula do not exist, the models did predict the present-day occurrence of the species in the regions of the Gulf of Biscay, the English Channel, and the North Sea. Regarding the future projections, there was a large northward shift of the centroid (Figure 1), due to the considerable decrease in the mean habitat suitability over most tropical and temperate latitudes for this species, concomitant with an increase in the mean habitat suitability at the higher latitudes of the Northern Hemisphere, in a trend that was increasingly exacerbated with the severity of the RCP scenario (Figure 6). The projected distribution exhibited a severe contraction over most of its predicted present-day distribution, with an expansion in the North Sea and the Arctic Circle (Figure 7). Indeed, it is possible that *G. catenatum* might expand northward in the Northern Hemisphere. There is already evidence of a poleward shift in the distribution of this species in the last century [39]. The predicted loss of habitat suitability in lower latitudes could indicate that this species may lose the ability to colonize and persist in these regions, undergoing localized extinction in these areas and, once more, shifting towards the poles. The projected losses in distribution (Figure 7), however, suggest that *G. catenatum* could be under severe threat in the tropical and subtropical regions worldwide, due to climate change. 

As discussed, the present SDM analysis predicts a potential poleward distributional shift for these three PST-producing species until the end of the century. Several prior studies projecting the climate change biogeographical effects on phytoplankton species have indeed suggested similar phenomena. By examining a 60-year ocean basin-wide time series of phytoplankton samples from the North Atlantic Ocean, Chivers et al. [137] found that dinoflagellates closely follow the velocity of climate change, i.e., the rate of isotherm movement, supporting the prediction that marine phytoplankton will move poleward with rising temperatures [138]. The present study’s projections are consistent with this data, suggesting that the three PST-producing species in the present analysis will likely undergo a poleward shift in their distribution (mainly northward), as their range of optimum oceanic conditions shifts, and their survival and adaptability are put to the test at lower latitudes. This poleward shift in distribution is conducive to the potential development of HABs in regions such as the north-eastern Atlantic and the north-western Pacific Oceans, as was projected for other HAB genera, e.g., *Prorocentrum* spp. and *Karenia* spp. [139]. Likewise, other dinoflagellate genera, such as *Ceratium* spp. and *Dinophysis* spp., have also exhibited signs of poleward distributional shifts, the former expanding from their tropical ranges into the recently warmer waters in the temperate North Sea [140], and the latter on the Pacific coastlines of South America [141]. At the same time, evidence of these distributional changes also exists for other phytoplankton groups (e.g., coccolithophores [142]). However, HAB events result from a complex interplay of biotic and abiotic interactions, of which the habitat suitability is a prerequisite but not the sole player. The present results do not allow predicting the likelihood of future PST events but contribute for projecting species’ movements and the identification of risk areas. 

It is necessary to address some potential caveats of the SDM analysis. As mentioned before, the models were subject to some degree of over and/or underprediction in all the species. Overprediction is relatively common in this type of approach, since the SDMs assume the complete occupation of areas that are climatically suitable to a given species [143]. As such, the models will predict a species to occupy a given geographical area, despite the species not occurring there due to other unaccounted environmental predictors, geographical barriers [144], etc. At the same time, the models fail to predict the presence of a given species (i.e., underprediction) in specific areas where there are historical records of its occurrence. This issue is directly linked to the nature of the occurrence point data. Occurrence data is mainly retrieved from online databases, such as OBIS (i.e., the Ocean and Biodiversity Information System database) and GBIF, and these are likely unable to produce a complete inventory of a species’ known distribution. The data from these databases are dependent on the sampling effort, which is usually geographically biased [145], due to reasons such as the accessibility and funding of directed scientific surveys. This can explain, for instance, for *A. minutum*, the absence of a presence prediction in certain regions, suggesting that the presence data included did not accurately describe the complete range of the environmental space occupied by this species. Notwithstanding, this caveat does not diminish the predictive potential of the models in suggesting the overall global trends of future poleward shifts in the species’ distribution, together with the decreasing habitat suitability, and consequent potential loss in its distribution at lower latitudes. 

Another important issue refers to the inability of long-term SDM projections to model seasonal changes in abundance in each species. This is a potential limitation, as the simple occurrence of a species does not necessarily produce an algal bloom. However, the present study suggests an overall shift in habitat suitability, followed by changes in the distribution ranges of the three species modelled into areas where they previously did not occur. These shifts pose a serious ecosystem risk, as they open the possibility of nuisance blooms in novel areas [139]. Further research should also consider the contribution of higher trophic levels, which could alter, to some extent, the relative impact of these projections. Indeed, phytoplankton are confronted with strong predation pressure by protozoan and metazoan grazers [146]. By projecting the distribution of grazer species under the same RCP scenarios, and to the same time-frames, the models could use this ecological pressure to produce more accurate estimates. This could, for instance, be achieved by developing trophic interaction distribution models (TIDM) [147].

## 5. Final Considerations

Anthropogenic climate change is driving significant shifts in the distribution of phytoplankton species worldwide, causing major reshuffles in these marine communities [70,138]. Understanding how HAB species (particularly the toxin-producing ones) will potentially respond to the future marine climate change is of paramount importance, given the potential of these species to severely affect both the environment and human health alike. The present study employed a species distribution model framework to project the potential changes in the occurrence and distribution of three PST-producing species under four RCP scenarios, with the results suggesting an overall tendency for an increasing habitat suitability at higher latitudes (particularly in the Northern Hemisphere) and a decreasing suitability in tropical areas. The projected distribution of each studied species closely followed this moving suitability trend over both geographical space and time (i.e., poleward shifts), and the effects became exacerbated with the increasing RCP severity. Notwithstanding, further research is required to gain a more holistic understanding of the potential impacts of climate change on PST-producing species. First, the inclusion of a more varied assemblage of environmental predictors, e.g., variables such as nutrient input, pH, and light, could further improve the ecological relevance of the SDM projections. At the same time, the inclusion of more data points in the occurrence dataset, particularly in areas of the environmental space that are poorly sampled, would allow the models to better describe the relationships between the environmental predictors and the habitat suitability for each species, thus improving the model projections [148]. Finally, these results should be complemented with data regarding potential geographical changes in bloom event-likelihood for each species, to identify not only new potential areas of occurrence but also quantify the increased HAB risk. Notwithstanding these limitations, these SDMs can provide a general indication of the future trends in habitat suitability and species distribution, which can be of extreme importance to support sustainable ocean management and spatial planning processes [40]. 

## Figures and Tables

**Figure 1 biology-11-01424-f001:**
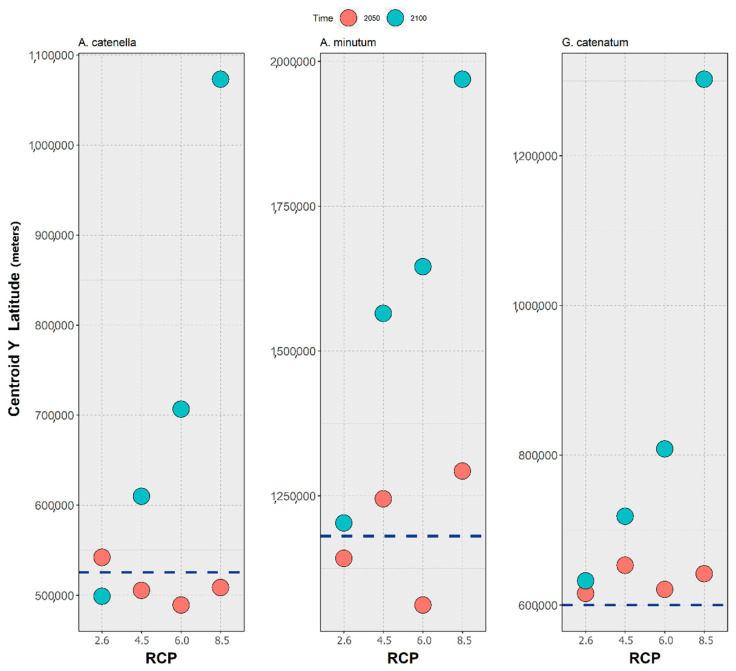
Changes in the centroid of latitudinal distribution (i.e., the mean latitude of the occupied pixels) calculated for 2050 (red circles) and 2100 (blue circles) and Representative Concentration Pathway scenario (RCP-2.6, 4.5, 6.0, and 8.5; CMIP5) for each of the three PST-producing HAB species (*A. catenella*—left, *A. minutum*—centre, and *G. catenatum*—right). The dark blue, dashed horizontal lines represents the centroid value for the present-day (2000–2014 environmental conditions based on monthly averages).

**Figure 2 biology-11-01424-f002:**
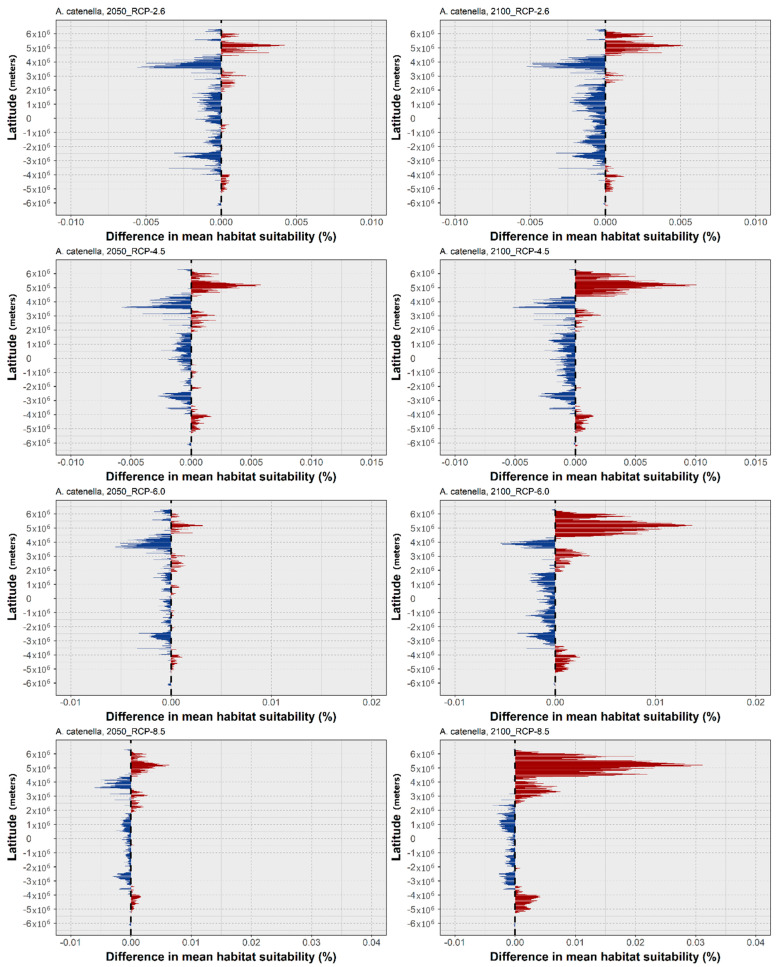
Projected changes in mean latitudinal habitat suitability for *A. catenella* between the present-day and 2050 (**left side**) and 2100 (**right side**), for each Representative Concentration Pathway scenario (RCP-2.6, 4.5, 6.0, 8.5; CMIP5), relative to the present-day baseline (i.e., the black dashed vertical line). Data to the right of the dashed line represents gains (red) and data to the left represents losses in mean habitat suitability (blue).

**Figure 3 biology-11-01424-f003:**
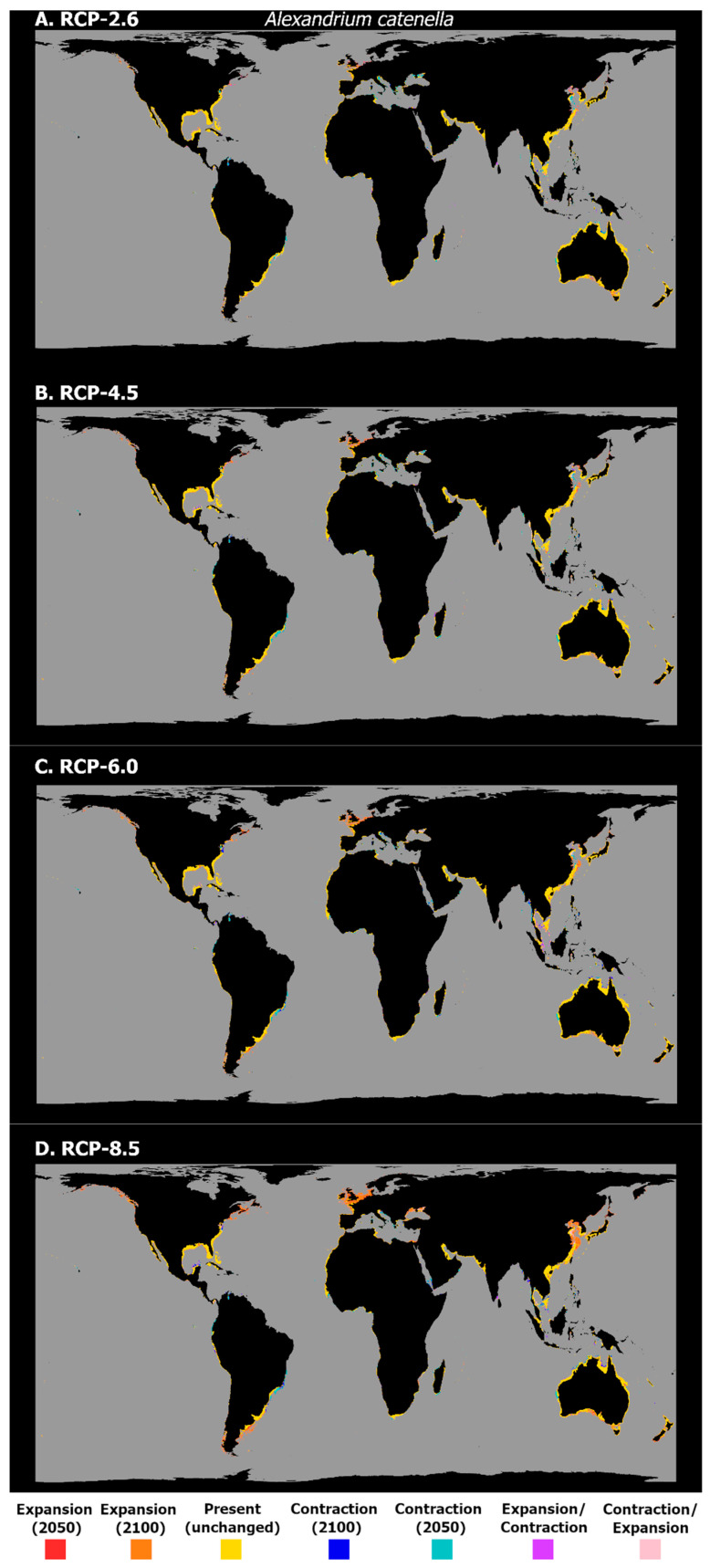
Projected distribution time maps for *A. catenella*, showing predicted unidirectional range shifts in occurrence distribution [i.e., range expansion (in red and orange) or contraction (in dark and light blue)] as well as transitory fluctuations (i.e., range contraction followed by expansion (pink), and vice-versa (purple)), for each Representative Concentration Pathway scenario (RCP-2.6, 4.5, 6.0, 8.5; CMIP5). A larger, higher-quality version of this figure is present in the Appendix A (See “High_Res_Figures” folder).

**Figure 4 biology-11-01424-f004:**
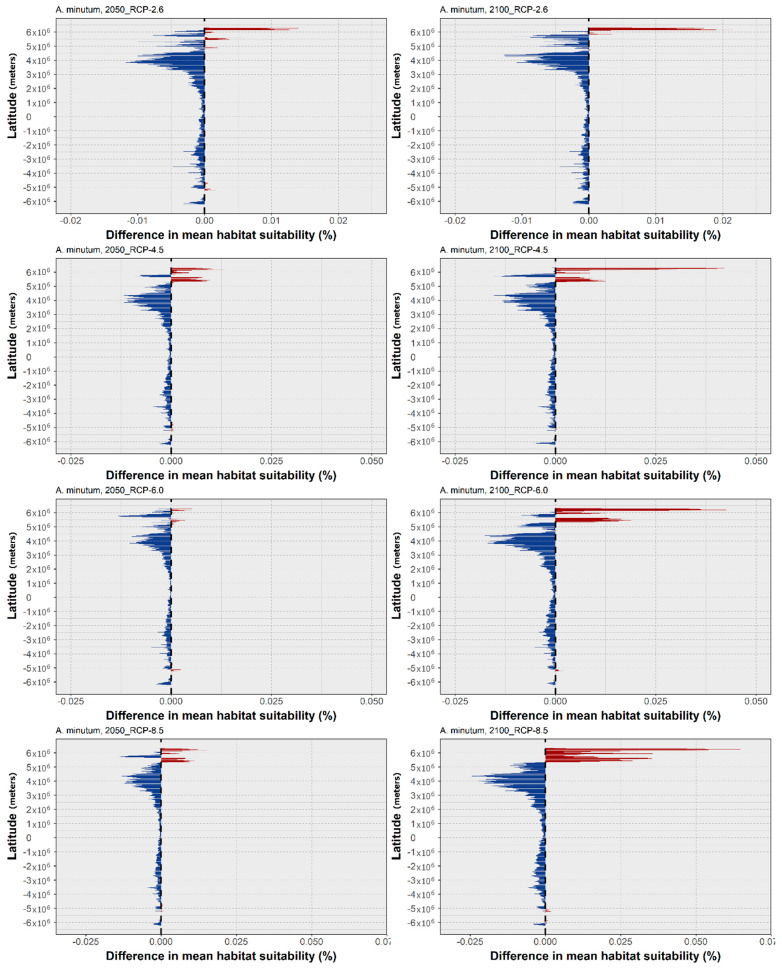
Projected changes in mean latitudinal habitat suitability for *A. minutum* between the present-day and 2050 (**left side**) and 2100 (**right side**), for each Representative Concentration Pathway scenario (RCP-2.6, 4.5, 6.0, 8.5; CMIP5), relative to the present-day baseline (i.e., the black dashed vertical line). Data to the right of the dashed line represents gains (red) and data to the left represents losses in mean habitat suitability (blue).

**Figure 5 biology-11-01424-f005:**
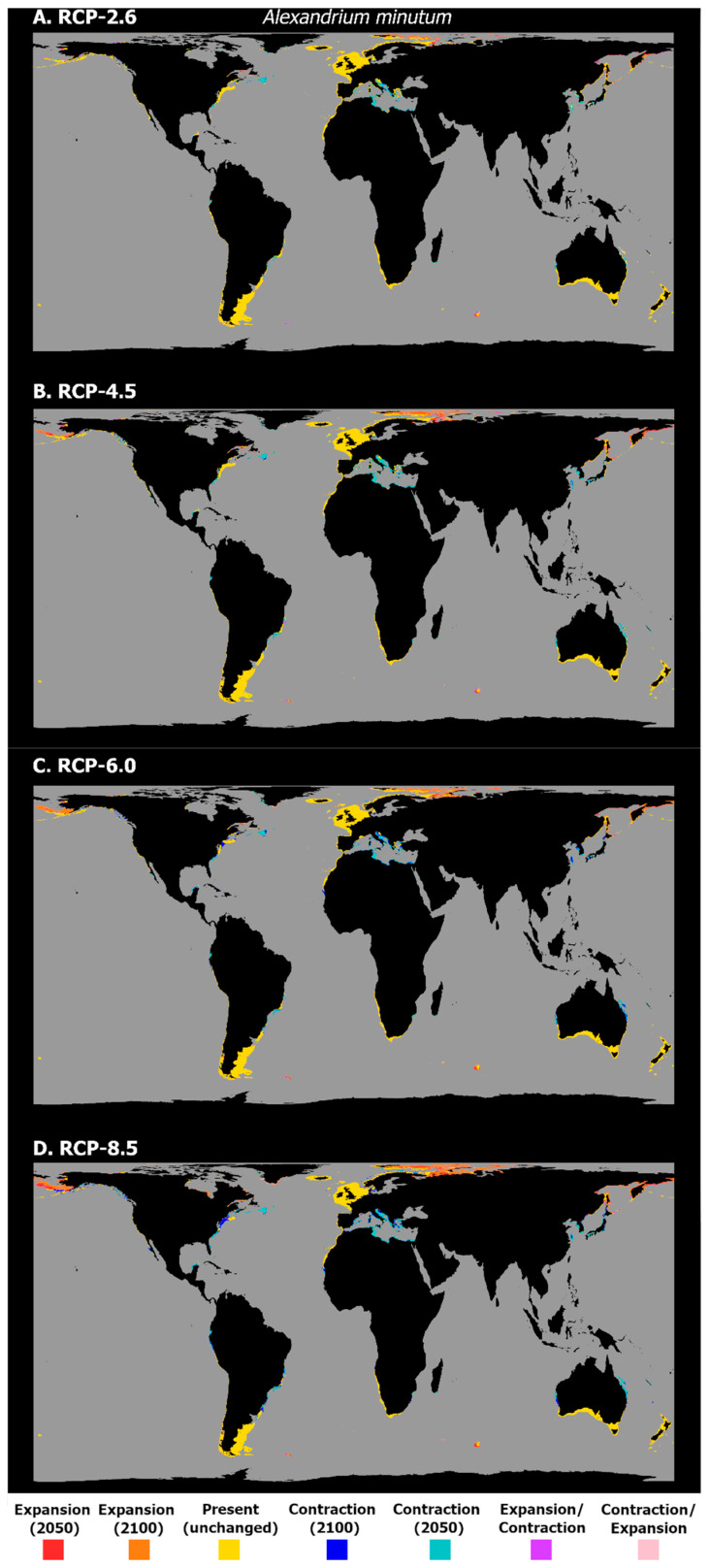
Projected distribution time maps for *A. minutum*, showing predicted unidirectional range shifts in occurrence distribution [i.e., range expansion (in red and orange) or contraction (in dark and light blue)] as well as transitory fluctuations (i.e., range contraction followed by expansion (pink), and vice-versa (purple)), for each Representative Concentration Pathway scenario (RCP-2.6, 4.5, 6.0, 8.5; CMIP5). A larger, higher-quality version of this figure is present in the Appendix A (See “High_Res_Figures” folder).

**Figure 6 biology-11-01424-f006:**
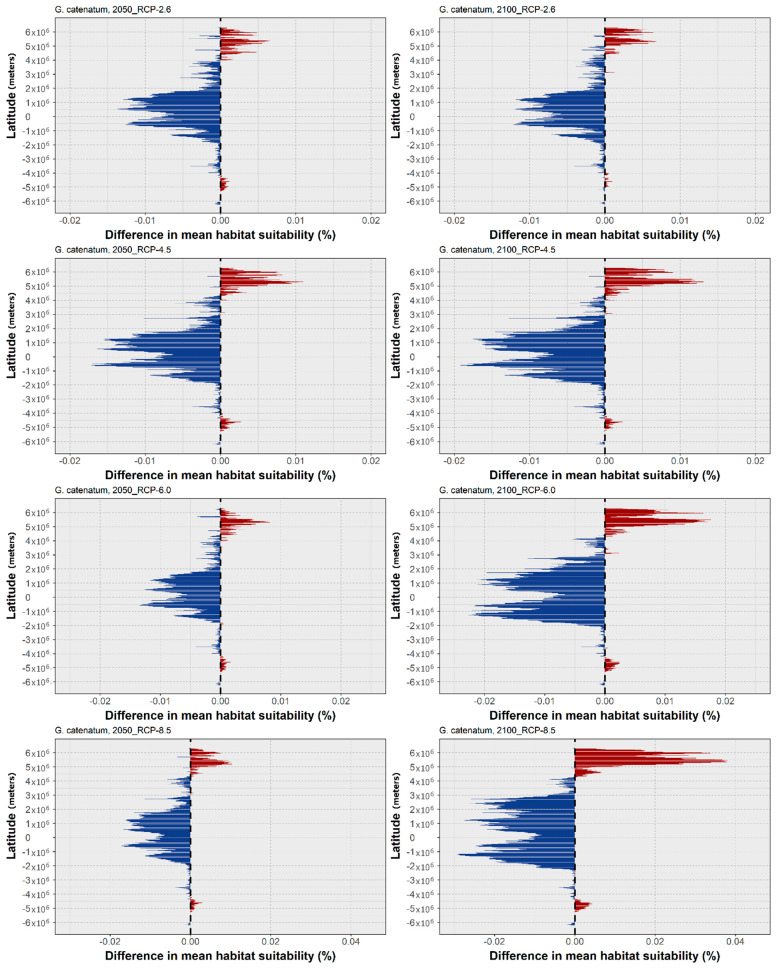
Projected changes in mean latitudinal habitat suitability for *G. catenatum* between the present-day and 2050 (**left side**) and 2100 (**right side**), for each Representative Concentration Pathway scenario (RCP-2.6, 4.5, 6.0, 8.5; CMIP5), relative to the present-day baseline (i.e., the black dashed vertical line). Data to the right of the dashed line represents gains (red) and data to the left represents losses in mean habitat suitability (blue).

**Figure 7 biology-11-01424-f007:**
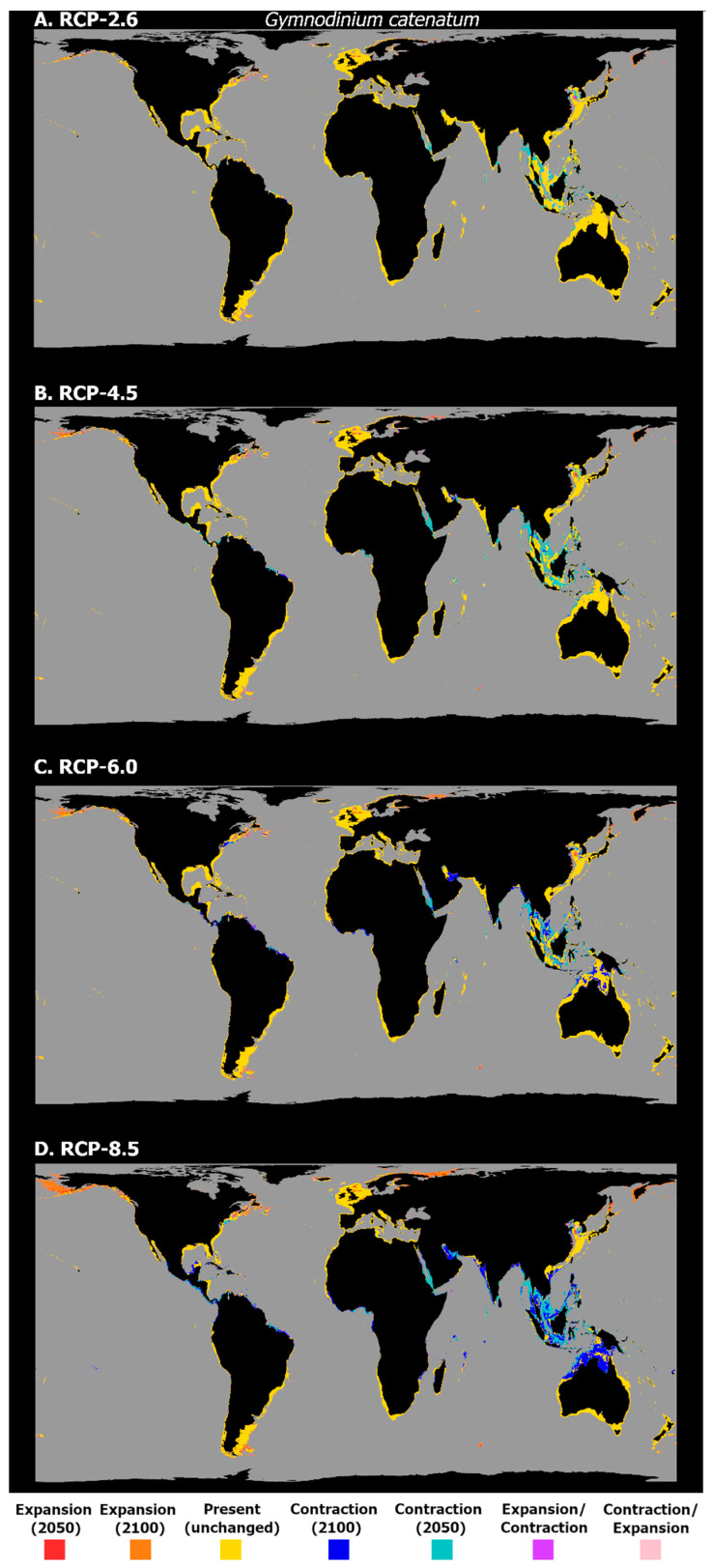
Projected distribution time maps for *G. catenatum*, showing predicted unidirectional range shifts in occurrence distribution [i.e., range expansion (in red and orange) or contraction (in dark and light blue)] as well as transitory fluctuations (i.e., range contraction followed by expansion (pink), and vice-versa (purple)), for each Representative Concentration Pathway scenario (RCP-2.6, 4.5, 6.0, 8.5; CMIP5). A larger, higher-quality version of this figure is present in the Appendix A (See “High_Res_Figures” folder).

**Table 1 biology-11-01424-t001:** Pre- and post-curation number of valid entries per species.

Species	Pre-Curation	Post-Curation
*Alexandrium catenella*	113	108
*Alexandrium minutum*	205	181
*Gymnodinium catenatum*	244	116

## Data Availability

The data has been made available in the Appendix A.

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
