# Peer review of "Projecting Future Climate Change-Mediated Impacts in Three Paralytic Shellfish Toxins-Producing Dinoflagellate Species"

_biology, 2022, doi:10.3390/biology11101424_

Round 1
Reviewer 1 Report
Borges et al. use Species Distribution Modelling to predict future changes in the abundance of three dinoflagellate species that can cause paralytic shellfish poisoning. I am not an expert in modelling, so I won’t comment of the methodological details. The topic is interesting and timely. I have some suggestions to improve the manuscript, especially the discussion which in my opinion does not address several important aspects.
Line 86: “blooms” not in capital
Line 87: replace “single” with “only”
Line 109: delete “most”
I am not an expert in modelling so I can’t say anything about the methods part of the model. I hope that the other reviewers have experience with modelling and can give advice here.
Line296-299: I find this unnecessarily complicated, maybe change to “…net difference between present-day and future habitat suitability indicates a considerable greater loss in 2050, when compared to the year 2100…”
Figures 2-4: Please increase the size of the axis labels.
I am missing a critical discussion of the data. The authors only summarise where the model predicts the species to be and if this matches actual reports of the species. However, there is no discussion about the environmental relevance of the data. The simple abundance of a potentially toxic species does not necessarily have a significant effect on the ecosystem unless the start to bloom. I understand that the model might not be able to predict seasonal changes in abundance of each species but this limitation needs to be discussed. Similarly, the impact of grazing in phytoplankton abundance is not considered in the model and not discussed.
Also, I miss an explanation/discussion about the size of the predicted change (presence and habitat suitability). Is a change in habitat suitability between 0.02 and 0.86% relevant at all? Similarly, how relevant are global percent changes between 0.0003 and 0.013%? Wouldn’t it be more important to look at changes in the likelihood of mass blooms of these species rather than general abundance?
I am further missing a discussion about the difference between the 2050 and the 2100 predictions. Why is habitat loss so much greater in 2050 compared to today and then improves again towards the end of the century? What causes these differences and is this trend expected to continue?
Reviewer 2 Report
The paper looks at modelling the potential habitat changes of three globally distributed toxin producing dinoflagellate species within this century. They apply a species distribution model (MaxEnt) to link the curated distribution records of each species with the underlying environmental conditions. The present habitat suitability is modelled using the MaxEnt model and projected changes are quantified by predicting the habitat suitability under future environmental conditions using several climate change scenarios. Given the importance of HABs and their effects on coastal environments it is important to understand how the future warming of our oceans will alter the distribution of these species.
The methodology is concise, detailed with specific attention brought to the many inherent biases that are encountered in species distribution modelling. However, I feel the results need to be expanded on beyond a simple number that highlights a net global increase or decrease in habitat suitability. There are also details from the MaxEnt models which are not included here but are in the supplementary. They are quite important in interpreting the changes which I expand on below.
General Comments on Results:
Please report your differences in percentages (0-100%) or as a proportion between 0 and 1. It looks like all net differences are less than 1% instead of what is intended.
I think you’re hiding a lot of detail by just reporting a single number for net difference. While discussed in terms of habitat gain/loss and contraction we don’t see the numbers that show where spatially that the net difference is being driven by. For example would it be possible to view latitudinally where the gain/loss in the cells are. E.g. – see attached pdf
I’m trying to understand the difference between net change and percentage change. I think the paragraph between lines 279-293 needs to be simplified to help that.
The figure legend is also confusing. You mention that in the left panels the colours represent range expansion or contraction but also have an inset legend that suggests red and blue represent the different time periods.
I have many questions about the percentage change - Is percentage change the perceived rate of change over time which is why the numbers are so small? If they are not rate of change then what do they represent.
MaxEnt results : Exploring the supplementary information I notice that the importance of your variables are skewed heavily towards the bathymetry (60%+). The dynamic environmental variables have a very small role to play. That doesn’t necessarily mean that they are not important but it is still important that the readers are made aware of this.
In the discussion the authors mention poleward expansion of species. Could this be quantified? Could I suggest something similar from this paper where the change in central distribution is calculated from the habitat suitability values. While this was done across 2 dimensions you could imagine a simplified version just examining the latitude dimension.
https://doi.org/10.1073/pnas.1519080113
Other comments
While I have some specific comments in places I feel it is best to just focus on the major structural comments right now.
Lines 109-118: This detail belongs in the method section.
Lines 295-328: Will hopefully be helped with the changes suggested above particularly regarding the use of % with what I believe are values scaled between 0 and 1.
The remainder of the results separate the results of the projected species change for each species. The results are a little heavy here and feel like just an endless list of coastal regions. I would try and summarise this by highlighting common themes of expansion/contraction among species and identify certain key areas.

Round 2
Reviewer 1 Report
I am happy with these changes
Author Response
The authors are glad to have successfully addressed all the Reviewer's questions and suggestions and thank the Reviewer again for his/her time and help in improving the quality of the manuscript.
Best regards,
Francisco O. Borges
Reviewer 2 Report
I'm quite pleased with the improvements made to the manuscript. Much of the figures have been drastically improved and the findings are now much more clear to follow.
Author Response

(The authors gave the same response as above.)
